

# Fuzzy-based optimization of AODV routing for efficient route in wireless mesh networks

Ibrahim Alameri[1,2,3], Jitka Komarkova[3] and Tawfik Al-Hadhrami[2]

[1] Institute of System Engineering and Informatics, University of Pardubice, Pardubice, Pardubice, Czech Republic
[2] Computer Science Department, School of Science and Technology, Nottingham Trent University, Nottingham, United Kingdom
[3] Computer Science, Jabir ibn Hayyan Medical University, Najaf, Najaf, Iraq

## ABSTRACT

The performance of any communication system heavily relies on the efficient routing of interventions. This article addresses the significant issue of routing protocol selection for optimal path determination in networks. Particularly, when wireless communication occurs among mobile nodes with limited resources, such as batteries, the routing problem becomes even more challenging. This article proposes the Fuzzy Control Energy Efficient (FCEE) routing protocol to overcome these challenges. The FCEE protocol combines the Ad-Hoc On-Demand Distance Vector (AODV) protocol with fuzzy logic techniques to enhance network lifetime and performance. The proposed approach introduces a memory-based channel integrated with fuzzy logic methodologies, which effectively restricts the forwarding of unnecessary broadcast packets based on the energy availability of the operating node. Through extensive simulations, demonstrate the promising capabilities of FCEE as a routing protocol for wireless mesh networks. To further assess the effectiveness of the FCEE protocol, the proposed article compares it with two existing routing protocols: AODV and Intelligent Routing AODV (IRAODV). The simulation results shows that the FCEE routing protocol significantly enhances the reliability of the conventional AODV, providing improved link connectivity and longer route lifetimes. Additionally, our proposed protocol enhances the quality of service (QoS) for mesh routing, with an average throughput of 351.374 (Kbps) compared to 90 (Kbps) for IRAODV and 39 (Kbps) for AODV. Moreover, FCEE exhibits superior energy efficiency with an average energy consumption of 14, while IRAODV and AODV consume 40 and 90 joules, respectively. In conclusion, the FCEE routing protocol demonstrates its potential to address the challenges of efficient routing in wireless mesh networks. By leveraging fuzzy logic and integrating it with AODV, FCEE enhances network lifetime, performance, and energy efficiency, making it a promising solution for future wireless communication systems.

Corresponding authors
Ibrahim Alameri, st61833@upce.cz
Tawfik Al-Hadhrami,
tawfik.al-hadhrami@ntu.ac.uk

## INTRODUCTION

In today's technological landscape, wireless networks have gained widespread usage owing to their remarkable mobility, flexibility, cost-effectiveness, and ease of deployment. The IEEE LAN/WAN standard committee's working group (WG 11) has successfully developed a wireless local area network (WLAN) based on the IEEE 802.11 family of specifications, solidifying its position as a prominent wireless network type. Alongside WLAN, other prevalent wireless networks include metropolitan area networks (MAN), vast area networks (WAN), cellular networks, and mesh wireless networks, the latter being particularly noteworthy due to its absence of wired infrastructure. Mesh networks, consisting of interconnected nodes without a centralized administration or infrastructure, have emerged as a distinctive solution for overcoming such limitations (*Abbas, Ilkan & Ozen, 2015*).

The term "mesh networking" encompasses various network paradigms, such as mobile ad-hoc networks (MANETs), wireless sensor networks (WSNs), vehicular ad-hoc networks (VANETs), airborne networks, underwater networks, personal area networks, home networks, and more. These versatile networks exhibit immense potential for utilization in diverse domains, including commercial, military, and civilian applications.

Mesh networks, characterized by multi-hop communication and predominantly wireless node connectivity, feature highly dynamic connections and routes that adapt to network growth and evolution. The majority of traffic within mesh networks is peer-to-peer in nature. Alternatively, mesh networks can also consist of a combination of wireless, mobile, and fixed nodes, where the primary traffic flow is from user to gateway. In contrast, connections and routes in mesh wireless networks are less dynamic. These networks often serve as extensions to existing infrastructure networks when time and budget constraints prevent the installation of new cables.

Driven by escalating communication demands, wireless mesh networks (WMNs) have emerged as a byproduct of wireless network evolution, ensuring ease of access and a wide range of functionalities. WMNs offer numerous advantages over other wireless network types, including cellular networks and wireless sensor networks, particularly in terms of supporting diverse applications. WMNs exhibit traffic patterns akin to mesh networks, as they frequently involve stationary routers. Network industries have been particularly drawn to WMNs due to their exceptional features, such as multi-hop routing, self-configuration, self-healing, self-management, reliability, and scalability. These attributes contribute positively to WMNs, including cost-effectiveness, simplified maintenance, robustness, excellent service provision for various applications, and the ability to deliver optimal performance while expanding capacity to meet growing demands. WMNs typically exhibit three main architectural types based on node functionality: infrastructure or backbone WMNs, client WMNs, and hybrid WMNs combining both infrastructure and client meshing. While backbone WMNs are the most prevalent, client WMNs allow for peer-to-peer mesh networks without the need for mesh routers, enabling end-users to facilitate packet forwarding. Hybrid WMNs combine the strengths of both infrastructure

and client meshing, with network infrastructure comprising mesh router nodes and end-users functioning as mesh clients.

Moreover, clients and backbone nodes can forward packets to the destination within WMNs. Hybrid mesh networks are anticipated to be the preferred choice for the next generation of WMNs. Despite node statics, the dynamic nature of link qualities poses a complex routing problem within wireless mesh networks. Thus, developing an efficient routing mechanism capable of path determination based on specific performance metrics associated with link quality remains a significant challenge within WMNs.

The establishment of effective paths between source and destination nodes constitutes a primary routing challenge in WMNs. Consequently, selecting an optimal routing protocol becomes a crucial task in mesh networks, necessitating the ability to dynamically adapt routing policies to changing network conditions and routing information inaccuracies. Adaptability is a critical evaluation criterion, encompassing factors such as transportation, energy, and mobility.

Therefore, the selection of an optimal routing protocol represents a key challenge in mesh networks. It is imperative that a routing protocol for mesh networks possesses the capability to dynamically adapt its routing policy in response to evolving network conditions and routing information discrepancies. Specifically, adaptability should be evaluated in diverse contexts, including transportation, energy, and mobility. Within the computational intelligence (CI) discipline, numerous strategies have been proposed, with many adaptive routing protocols leveraging these techniques. However, due to the limited energy and computing resources within mesh networks, specific CI approaches such as reinforcement learning, fuzzy logic, and swarm intelligence are better suited. This study primarily focuses on the implementation of fuzzy logic within the context of mesh networks.

This work aims to comprehensively explore utilising fuzzy-logic-based routing protocols for mesh networks, particularly by introducing a dynamic membership function to enhance system adaptability. Network lifetime represents one of the most challenging issues within wireless networks, encompassing the duration during which a network remains operational and capable of fulfilling its designated tasks (*Thamizhmaran & Anitha, 2015*; *Pariselvam et al., 2021*; *Choudhury et al., 2015*). In wireless mesh networks, routing protocols play a pivotal role in enhancing network lifetime, as it directly depends on the energy consumption of network nodes. Researchers have dedicated their efforts to developing routing algorithms that minimize energy consumption while maintaining desirable throughput, packet delivery ratio, and low packet drop rates.

The primary objective of this proposed work is to develop an efficient routing protocol for wireless mesh networks that significantly reduces node energy consumption while simultaneously improving network lifetime, throughput, packet delivery ratio, packet drop rates, and end-to-end delay. To this end, a novel energy-efficient network routing protocol, named the Fuzzy Control Energy Efficient (FCEE) Routing Protocol for wireless mesh networks, is proposed.

The remainder of this article is structured as follows: "Literature Review", focuses on the literature review. The overview of AODV is shown in "Aodv Routing Protocol Overview",

and a summary of fuzzy logic in "A Brief Overview of Fuzzy Logic", includes fuzzy rules. "Proposed Architecture of Fuzzy Control Energy Efficient (FCEE) Routing Protocol", provides an overview of the proposed Fuzzy Control Energy Efficient (FCEE) Routing Protocol. "Simulation Model and Results", presents the simulation model and results of the work. "Conclusion", proposes the conclusion and future work.

## LITERATURE REVIEW

Several studies investigating AODV have been carried out on AODV modifications. In *Chandrasekaran & Selvaraj (2022)*, the major focus of this research is to satisfy and guarantee the quality of service (QoS) parameters and to identify the optimal and best feasible path that meets the required criteria in MANET. To achieve this, the authors have proposed a novel variant model based on differential evolution (DE) having a set objective function. They have minimized the set objective function in order to meet the QoS constraints. The proposed work claims to achieve better shortest-path discovery compared to the shortest-path algorithm of AODV. Furthermore, the model also achieves better efficiency in other QoS parameters required for the network. The main drawback of the research work is that they have not considered the end-to-end delay in the objective function while selecting the shortest path. The objective function computes only hop count, residual energy and the routing load which is not enough for calculating the shortest path. Furthermore, link failure or stability is also not considered after modification of AODV which may decrease the performance of the proposed work.

The authors aim in *Sarkar, Choudhury & Majumder (2021)*, was to improve the QoS of MANET in this research work by proposing a novel route selection technique that combines Ant Colony Optimization (ACO) with the AODV. The proposed work calculates the pheromone value of a route of the nodes along the path, considering multiple QoS parameters: congestion, several hops, end-to-end reliability, and residual energy. The proposed mechanism selects the path with the highest pheromone value to transmit the data, which is considered the most acceptable path for data transmission of the nodes in the network. The proposed protocol has been compared with several well-known protocols, including AODV, DSR, and E-DSR, in multiple and different simulation scenarios using Network Simulator (NS-2.35). The proposed schemes have considered multiple QoS parameters to find the best and optimal route from source to destination. However, they have yet to consider the essential parameters, such as the end-to-end delay, jitter, and energy consumption, especially those required to find the best route selection.

*Abbas, Ilkan & Ozen (2015)*, resented a fuzzy-based approach to improving the performance of the AODV routing protocol. The proposed work is implemented to create a path from source to destination by selecting the most trusted nodes in the network to accomplish this goal. The fuzzy inference system feeds parameters like node mobility, residual energy, and hop counts in the supposed work. The simulation results show that the approach proposed in this work outperforms the standard AODV and minimum batter cost routing protocols. The proposed work shows a good efficiency level. However, it may degrade its performance in some cases, such as the average end-to-end delay shown in work where the delay is higher in the case of an increase in pause time.

*Li et al. (2021)*, proposed a routing algorithm named Fuzzy-logic-assisted AODV (FL-AODV) to enhance the routing reliability in MANETs. The proposed algorithm selects the relay node based on the highest reliability to achieve this goal. The route is selected based on the accumulated reliability to be reserved for the transmission of data packets. The simulation results show that the proposed algorithm outperformed in terms of reliability compared to the mentioned protocols, AODV and FLRA. The proposed work claims to improve the reliability of the network. The claim is valid in the case of AODV. However, the algorithm is less reliable than FLRA according to the resulting value mentioned in the research work. Reliability contains multiple parameters to be improved, known as reliability improvement. Their work shows that some parameters are not improved, such as average reliability and average end-to-end delay.

*Perumal, Prabhu & Selvi (2022)*, proposed an improved priority-aware mechanism to provide enhanced quality QoS parameters, reduced end-to-end delay and consistent throughput in the network. As in previous works, it is an issue in both connection-oriented and connection-less approaches. In connection-oriented, it degrades the QoS of throughput and end-to-end delay when the speed of nodes exceeds two m/s, and in connection-less, it provides reduced QoS. However, they have yet to consider some essential parameters, such as energy consumption and link breakages, as they deal with mobility and avoid node speed. Furthermore, the simulation results in their work show that the throughput of AODV is better than the proposed protocol.

In *Priyambodo, Wijayanto & Gitakarma (2020)*, the authors evaluate the route request parameters (RREQ_RETRIES and MAX_RREQ_TIMEOUT) of the AODV protocol. The proposed study is only changing the combination values of the parameters as mentioned earlier and comparing AODV with OLSR routing protocols to select the best one for optimal data transmission over MANET. The authors claim that the reduction of combination values improves the quality-of-service parameters. However, there are not any significant improvements shown in the work. The figures show that the default AODV performs better with its default values.

A new routing protocol, Intelligent Routing AODV (IRAODV), is proposed in the work of *Anand & Sasikala (2019)*. In this work, the routing strategy establishes routes only when data transfer is required, as determined by the arrival of packets. One benefit is that it reduces the excessive routing overhead typical of proactive routing. In particular, it improves the efficiency of bandwidth use by decreasing the computational cost of maintaining and reestablishing routes at regular intervals and the frequency with which routing messages are exchanged among nodes. A drawback, however, is the lengthened time required at the outset to build pathways.

In *Er-rouidi et al. (2019)*, the authors proposed an energy-efficient AODV-based routing protocol known as EE-AODV. As opposed to merely considering the present residual energy of a node, this protocol considers the rate at which energy is being spent during each period. Previously, the only thing that was taken into account was this energy. EE-AODV can collect accurate information about the amount of energy utilized during the sending and receiving of packets by measuring the energy consumption rate. The complicated calculation of these quantities is optional to accomplish this goal. EE-AODV

is able to produce a more accurate estimate of the remaining lifetime of nodes by using data such as the residual energy and the estimated consumption rate. It was demonstrated that the EE-AODV dramatically reduces the energy the nodes consume compared to the standard AODV and the EQ-AODV (Energy and QoS-supported AODV), which were both used as points of comparison.

In *Bamhdi (2020)*, the author suggested an alternate protocol, dubbed Dynamic Power-AODV (DP-AODV). DP-AODV adapts the AODV protocol to adjust transmission power usage dynamically. To achieve this improvement, the DP-AODV protocol uses the dependence of a transmission range on density. Simulation results demonstrated that, as density increases, DP-AODV decreases delay, compared to AODV, and offers better performance for highly-populated networks exceeding 200 nodes. The simulation results demonstrated that DP-AODV improves network throughput while decreasing node interference in a dense area and that it also improves overall network performance in terms of increasing the fraction of delivered packets, decreasing control overheads and jitter, increasing overall throughput, decreasing interferences, and decreasing the end-to-end delay in medium-to-high density conditions.

In recent years, several pieces of research have been carried out on energy optimization and quality of service (QoS) in mobile ad-hoc networks (MANETs), as shown in the above studies. Most of these studies have focused on reducing the energy consumption of individual nodes by adjusting their transmission power or reducing the number of control messages exchanged. On the other hand, the related works have a limited QoS because balancing energy consumption and QoS is only sometimes possible because of the trade-off between these two parameters, which could lead to a limited QoS. The proposed work of FCEE focused on enhancing QoS and energy consumption. The work addresses this limitation by proposing an FCEE optimization approach that considers the fuzzy rules to improve energy efficiency and QoS, as shown by the simulation results.

The proposed approach differs from previous studies in that it also considers the nodes' energy level, which is critical for prolonging the network's lifetime. Simulation results show that the proposed approach outperforms existing energy efficiency and network lifetime methods. In conclusion, the FCEE routing protocol has scalability as an energy-efficient reactive routing system and enhances the QoS.

## AODV ROUTING PROTOCOL OVERVIEW

The routing protocol decides how the routers communicate to deliver information, allowing them to choose routes among any two nodes in a computer network. For mesh networks, the AODV routing protocol eliminates the problem of routing loops (*Abbas et al., 2020*). They can function in a network with multiple mobile nodes even though they have not been prompted to do so, and they are resilient against disruptions in the form of packet loss, link failure, and node mobility (*Taterh, Meena & Paliwal, 2020*; *Alameri et al., 2022*).

The routing protocol ensures that every node consistently updates its routing table. In the routing table, you can find the next hop node, the sequence number, and the total number of hops. You can calculate your current distance from the starting node to the final

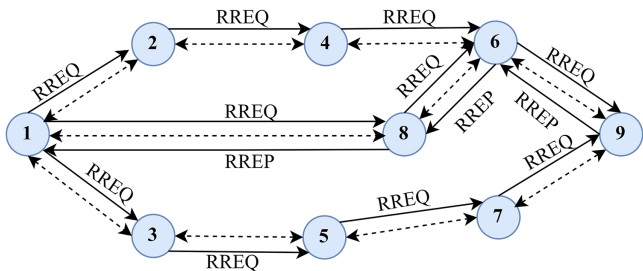

**Figure 1  Typical routing methodology in AODV.**

one by counting the number of hops. The node will forward all incoming data packets to the node after it. A time stamp in the sequence number indicates how recently the route was created (*Goyal, Sharma & Kumar, 2020*; *Goyal, Goyal & Kumar, 2020*).

The AODV protocol allows mobile nodes to acquire routes to fetch new destinations quickly and eliminates the maintenance of routes to destinations that are not currently active. It will enable mobile nodes to periodically acknowledge if a change in network topology or a link failure is detected (*Chaturvedi & Kumar, 2021*). The AODV protocol notifies the affected nodes during a link failure so that they can reject routes that use the failed connection. According to the AODV protocol, the destination is permitted to generate the destination sequence number, which is appended to any route information the destination sends to its requesting nodes (*Jatti & Kishor Sonti, 2022*).

## AODV routing methodology

As its name implies, AODV is a hop-by-hop routing protocol. Figure 1, illustrates the typical routing approach the AODV protocol takes. The source node sends the RREQ to the network to discover the path to the destination. The relay nodes forward the RREQ and set up the alternative route to the destination after receiving the request (*Hanan, 2020*). After receiving a route request, the node that knows the destination route sends back an acknowledgement called a route reply (RREP). The RREP has an adequate number of hops to get the user to where they need to go. The nodes must confirm *via* RREP that the source node has indeed set up a forwarding route to the destination (*Soomro et al., 2020*; *Sayedahmed, Fahmy & Hefny, 2020*). Thus, AODV uses a hop-by-hop routing approach to determine a path between a starting and end point.

It is possible to split the AODV routing mechanism's overall procedure into two distinct phases:

### Discovery of routes

Due to the legal route message from source to destination in its routing table, a source node will send a packet to a specific destination node during route discovery in WMNs to confirm with the routing table whether it contains the present route to the node. If the destination IP address is found in the node's routing table, the node sends the packet to the next hop node in the path. To begin a presentation towards the chosen destination node, the source node will engage in route discovery if it still needs to learn the path to the destination (*Perkins, 2008*).

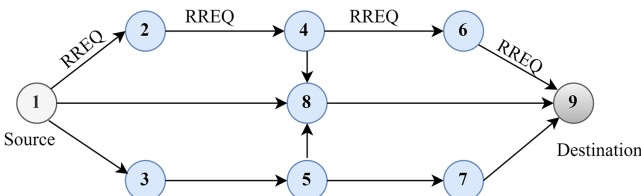

**Figure 2  Route discovery in AODV.**               

The node generates an RREQ packet with the destination IP address and the destination node's previous familiar sequence number to complete the route discovery process. Each time the source node generates an RREQ packet, the broadcasting ID included in the packet will increase by one. After sending out an RREQ packet, the sending node sets a timer to await an acknowledgement.

The AODV route-finding procedure is depicted in Fig. 2. The node first checks the RREQ packet's source IP address and then broadcasts the identity pair after confirming its legitimacy. Each node must keep a log of the source node's broadcasting ID and IP address for as long as possible for every RREQ packet it receives. The node must know the way to the destination, as indicated by its routing table, to respond to the RREQ packet. The RREQ will also relay the destination's sequence number to prevent routing loops. This ensures that the returned route is up-to-date and not from an older network topology at the former intermediate node. If the node satisfies those criteria, it will respond by uni-casting the RREP packet back to the sending node.

## Maintenance of routes

After a path between a source and destination has been discovered, the mobility of individual nodes in a wireless network will only affect the paths along which those nodes participate. If the source node's location has changed during data forwarding, the node at the receiving end can start the route over *via* discovery and build a new path to the destination. The affected source nodes will receive an error message labelled RERR, which stands for route error. It is started by the upstream node and lists the currently inaccessible destinations because of the list link. The surrounding nodes will receive a RERR notification. As soon as they receive the RERR signal, neighbouring nodes will set the value of the node to infinity to indicate that the path is useless. If the source node receives RERR, it restarts route discovery.

Discovering the original path is depicted in Fig. 3. Two approaches can be used for route maintenance. The first method involves sending an RREQ message from the source node to the RREQ of the neighbouring node, complete with source and destination IP addresses, sequence numbers, and broadcasting IDs. The source node receives RREP messages from the destination node through an intermediary node. Locally, a broken link can be repaired by an intervening node. The RREQ message is relayed from the intermediary node to its final destination *via* its neighbouring node. An RREP message is sent from the destination to the intermediate node to confirm the RREQ and reestablish the path between the source and the final hop.

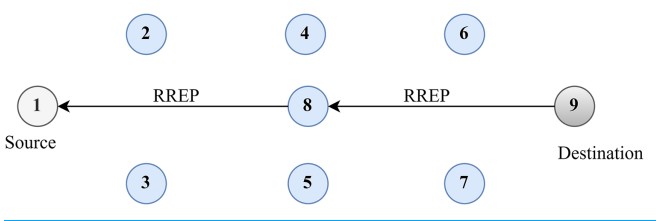

**Figure 3** Route discovery.               

# A BRIEF OVERVIEW OF FUZZY LOGIC

This research article designs and implements a new Fuzzy Control Energy Efficient (FCEE) routing protocol for wireless mesh networks. This section presents a brief refresher on fuzzy logic. Human beings can make decisions even when unclear or insufficient information is provided, but machines generally cannot. In 1965, *Zadeh, Klir & Yuan (1996)* came up with a new idea and coined the term "fuzzy logic".

Fuzzy itself means "vague" or "unclear". It is a method that allows one to incorporate human-like thought processes into a control system. Fuzzy logic has a low processing cost and excels at handling nonlinear, time-varying, and uncertain systems.

## Fuzzy set theory

In classical set theory, an element may or may not be a member of a set. In fuzzy set theory, however, an element can be a member of a set upto a certain breath. The fuzzy set theory is an extension or generalization of the classical set theory which is also known as crisp set theory. A crisp set or classical set, X, is typically defined as a collection of limited, countable, or uncountable elements or objects *i.e.*, $x \in X$. Each element can either be a member of a set or not. However, in reality, elements' membership in sets is not binary (1 or 0). Fuzzy set theory (FST) seeks to express imprecise information. For example, in wireless mesh networks, 'low energy', 'high throughput,' etc., is difficult to represent in conventional (classical) set theory.

A membership function that quantifies the level of membership for each element defines a fuzzy set. A fuzzy set 'A', of a discourse universe, X, is defined as follows: $A = (x, \eta_A(x))|\forall x \in X$, where $\eta_A(x)$ is a membership function of x with respect to fuzzy set A. Figure 4, presents a membership function.

## Fuzzy reasoning

Fuzzy logic was developed to translate human reasoning into a formalised mathematical expression. Fuzzy logic uses 'linguistic variables' and 'inference rules' to establish an approximation of a proposition's truth value rather than relying solely on classical reasoning. In classical reasoning method a proposition can have only two states either true or false. A variable whose values are words or phrases in a natural or artificial language is referred to as linguistic context. Hedge words like "more," "many," "few", etc., as well as connectors like AND, OR, NOT, *etc.*, can be used by domain expert to construct language rules based on linguistic variables. An inference engine will make use of these principles to speed up approximate reasoning.

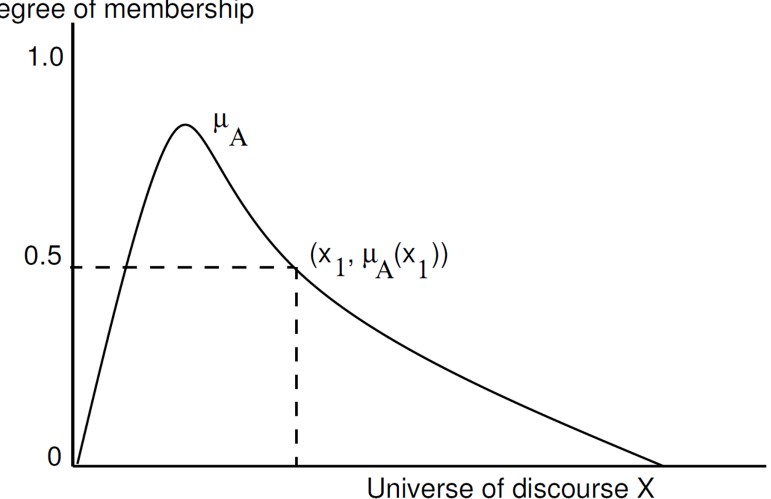

**Figure 4 Membership function of A.**

## Linguistic variables

It is a type of variable which derives its values from words or phrases. These words or phrases can belong to nature or artificial language. The main attributes are:

- It has a name.
- It has a set of values.
- Universe of Discourse; a set of all possible values.
- Syntactic Rules.
- Semantic Rules.

## Fuzzy rules

The core of fuzzy logic is based on its rules set, which is known as fuzzy rules. Fuzzy rules are basically "if and else" clauses. The establish relation between linguistic variables and possible outcome. Rules are generally built with help of domain experts.

## Fuzzy logic system

The fuzzy logic-based control system is represented in Fig. 5. It has following components.

- **Fuzzifier:** A fuzzier is a module that takes input data in its classical (crisp) form and transforms it into sets of data that are in their fuzzy form.
- **Knowledge base:** All fuzzy rules are stored in the knowledge-base.
- **Inference engine:** The inference engine handles the deliberation and selection of a course of action. The result of inference engine is fuzzy set.
- **Defuzzifier:** It converts fuzzy set back to classical/crisp values.

# PROPOSED ARCHITECTURE OF FUZZY CONTROL ENERGY EFFICIENT (FCEE) ROUTING PROTOCOL

The current section presents the proposed design for the Fuzzy Control Energy Efficient (FCEE) routing protocol for wireless mesh networks. The suggested work introduced a short-term memory approach in the fuzzy logic system which is presented in the subsection of the "Proposed short-term memory channel—based fuzzy logic system". Also, the proposed architecture is on top of the AODV routing protocol and would be present in the following subsection of the "Proposed short-term memory channel—based fuzzy logic system".

## Proposed short-term memory channel—based fuzzy logic system

The proposed work introduces a memory channel module in the fuzzy control system. Figure 5, presents the proposed new design. It includes a memory channel used to keep state variable(s), *i.e.*, last broadcast information. The inference engine uses this state information and the knowledge-based (fuzzy rules) to make a fuzzy output set. The fuzzy-output set is then fed into the defuzzifier module to produce the final output.

## Fuzzification

The fuzzifier module presented in Fig. 5, is used to perform the fuzzification process with the help of the membership function given in Eq. (1). This equation is used to determine the membership of the node. We defined four linguistic variables to express node's energy level *i.e.*, (i) high, (ii) good, (iii) average and (iv) low. Equation (1) is presented below and its graph is in Fig. 6.

$$\mu(x) = \begin{cases} 0 & \text{if } x \leq p \\ \frac{x-p}{r-p} & \text{if } p \leq x \leq r \\ 1 & \text{if } r \leq x \leq s \\ \frac{q-x}{q-s} & \text{if } s \leq x \leq q \\ 0 & \text{if } q \leq x \end{cases} \tag{1}$$

The scalar values *i.e.*, p, q, r, s are used to shape the function correctly. Scalar 'p' and 'q' are used to construct the base of the trapezoid while 'r' and 's' are used to construct the top of the trapezoid. Table 1, presented in the article is used to define the fuzzification process.

## Fuzzy rules

The inference engine needs information from the knowledge base for the proposed design. The knowledge base is mainly made up of fuzzy rules and memory channels. This section presents the fuzzy rules that are used by the model and discusses the memory channel.

**Memory channel:** The proposed model uses a memory channel to maintain the node's state information. The suggested work used the broadcast packet parameters as state information. Since this research aims to design a congestion-tolerant routing protocol, broadcast packets in any network affect network throughput and can become a leading source of congestion and packet loss.

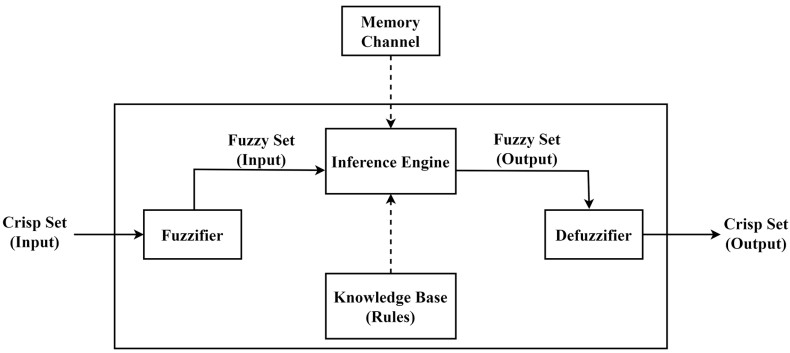

Figure 5 Proposed memory channel based FLS.     

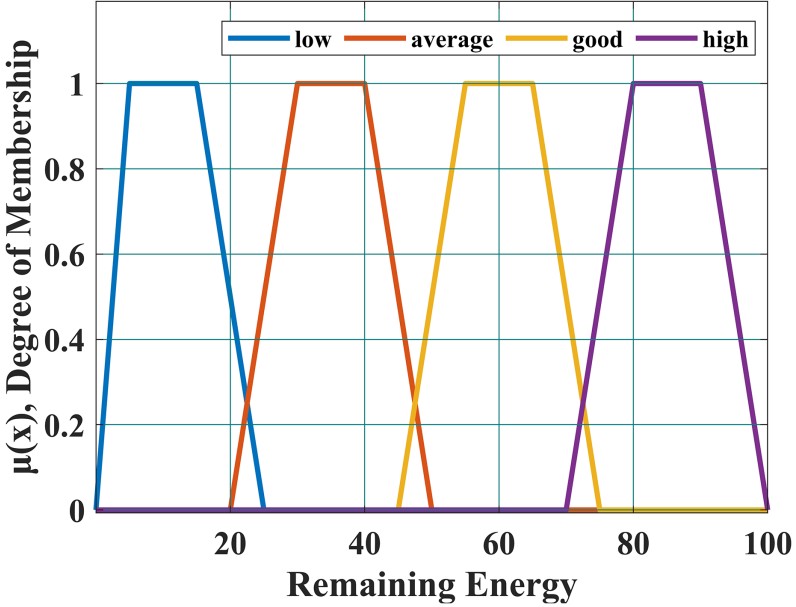

Figure 6 Membership function.     

| Table 1 Fuzzification. | |
| --- | --- |
| **Remaining energy (RE)** | **Energy level** |
| If a node has RE greater than 75%. | High |
| If a node has RE greater than (or equal to) 50% and less than 75% | Good |
| If a node has RE greater than (or equal to) 25% and less than 50% | Average |
| If a node has RE less than 25% | Low |

The proposed work used the node's remaining energy level and the last broadcast packet's information to decide whether the node would forward this broadcast packet or not. Fuzzy rules used by the inference engine are presented in Table 2.

**Table 2 Fuzzy rules used by proposed fuzzy system.**

| Rules | Energy level | Last broadcast | Decision (Forward broadcast) |
|-------|--------------|----------------|------------------------------|
| Rule 1 | High | Yes | Yes |
| Rule 2 | High | No | Yes |
| Rule 3 | Good | Yes | No |
| Rule 4 | Good | No | Yes |
| Rule 5 | Average | Yes | No |
| Rule 6 | Average | No | Yes |
| Rule 7 | Low | Yes | No |
| Rule 8 | Low | No | Yes |

## Inference engine

The inference engine will use fuzzy rules and memory channel state information to make a decision. The output generated by the inference engine is again a fuzzy set, which will be input to the defuzzifier module.

## Defuzzifier module

The Defuzzifier module will convert the fuzzy set generated by the inference engine back into a crisp set. Defuzzification is accomplished in this article using the centre of gravity (COG) approach. When using COG, the key principle is to output the membership function curve's centre of gravity and the enclosed area of its abscissa. The basic principle in the CoG method (*Subbotin, 2014*; *Subbotin & Voskoglou, 2014*; *Subbotin, 2015*; *Van Broekhoven & De Baets, 2006*) is to find the point X* where a vertical line would slice the aggregate into two equal masses. If $\mu_c$ is defined with discrete MF, as express in the Eq. (2).

$$x^* = \frac{\sum_{i=1}^{n} \mu_c(x_i) \cdot x_i}{\sum_{i=1}^{n} \mu_c(x_i)} \tag{2}$$

This method returns a precise value depending on the fuzzy set's centre of gravity. The overall area of the membership function distribution used to describe the combined control action is divided into a number of sub-areas (in our case, trapezoidal). Each subregion's area and centre of gravity, or centroid, are calculated. Then the sum of all these sub-areas is used to determine the defuzzified value for a discrete fuzzy set. Figure 7, presents the CoG method for defuzzification.

Let $B_i$ and $x_i$ denotes the area and center of gravity of ith sub-region, as shown in the Eq. (3).

$$x^* = \frac{\sum_{i=1}^{n} B_i \cdot x_i}{\sum_{i=1}^{n} B_i} \tag{3}$$

## FCEE flowchart

Congestion in the network actually occurs at the intermediate nodes. Intermediate nodes are also known as routing nodes, which make routing decisions. The goal of a router or

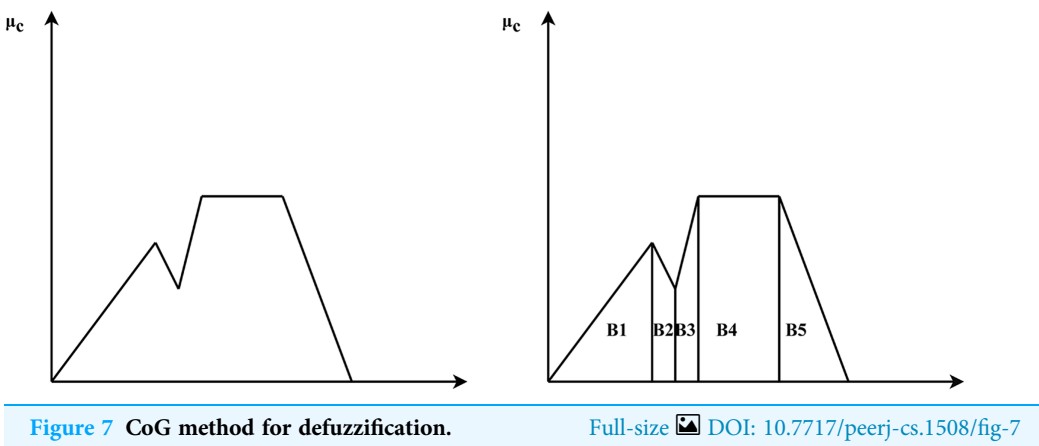

**Figure 7 CoG method for defuzzification.**

routing node is to select the best path and forward the packet towards the next hop or destination. Another leading source of congestion is broadcast traffic. Networks under high broadcast traffic are prone to congestion. The proposed Fuzzy Control Energy Efficient (FCEE) routing protocol exploits the fuzzy logic and memory channels to control the broadcast efficiently. Due to this, the network load is optimized in a better way, and the network becomes less congested. Figure 8, presents the FCEE algorithm's flow chart.

## Flowchart of the proposed model overview

The presented subsection explains the working of the FCEE flowchart illustrated in Fig. 8. Each step of the flowchart is explained below.

1. When a device, often termed a node, receives a packet from the wireless channel, it processes the packet within the network stack. The datalink layer plays a pivotal role in handling the packet at the link level, encompassing tasks such as error checking, correction, and packet synchronization. As the packet moves up the network stack, it reaches the network layer, known as Layer III. At this level, various tasks are executed in protocols like TCP/IP, including calculating and verifying the header checksum, validating the Time-To-Live (TTL) field, and routing the packet towards its intended destination. The subsequent layer, Layer IV or the transport layer, ensures the reliable transmission of the packet from source to destination. It achieves this through error detection mechanisms, such as checksums, prompting necessary actions like retransmissions.

2. After performing the routine operational tasks, the routing sub-system determines the destination of the node.

3. If the receiver (who received the packet) itself is the destination node, then the packet is delivered to the application process or the service.

4. However, if the packet is to be forwarded to other nodes on the network, then routing protocol FCEE will start its operation.

5. If the destination address of the node is a uni-cast address of a particular node, then the FCEE routing protocol will forward the packet towards the target node. But if the

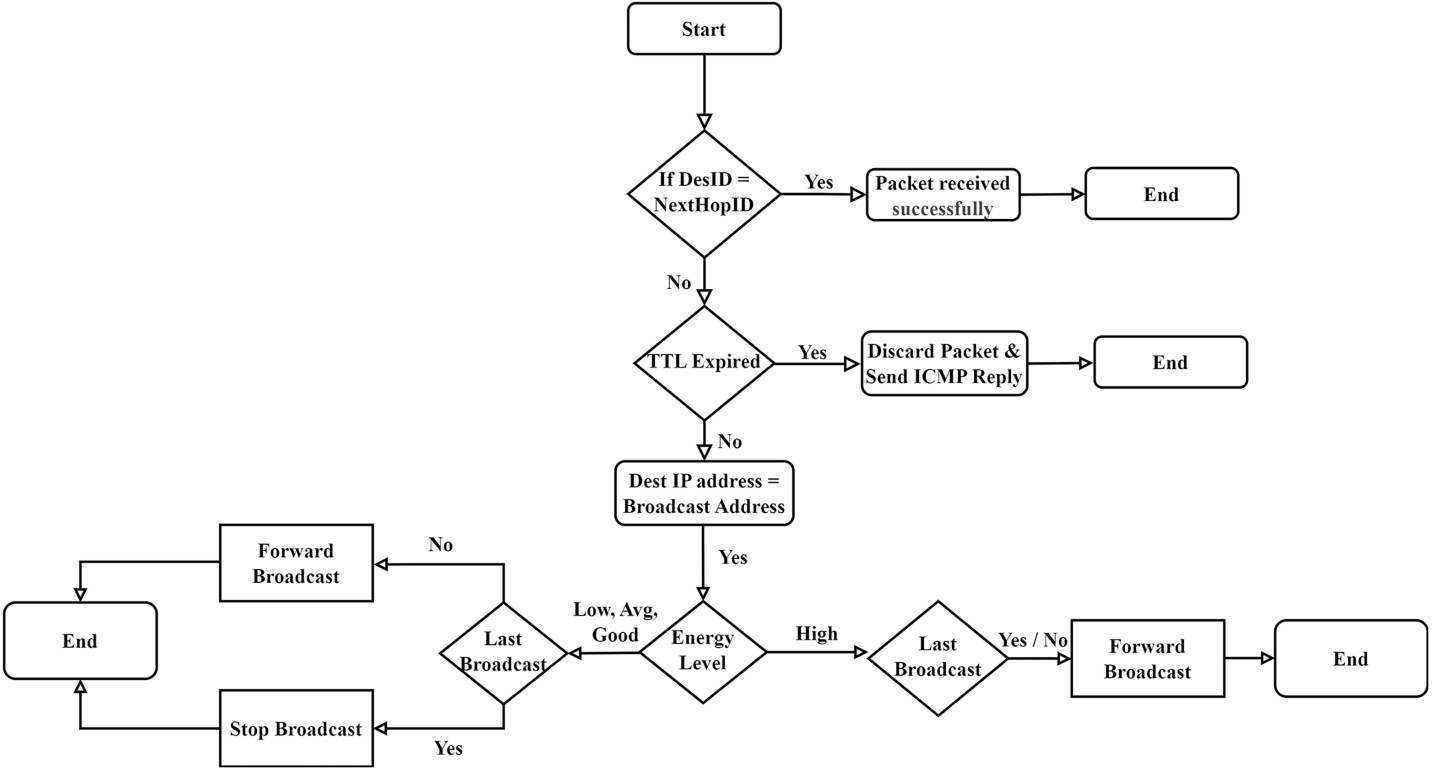

**Figure 8 Fuzzy control energy efficient (FCEE) algorithm flowchart.**

destination address is a "broadcast" address, the FCEE routing protocol will check its "content addressable memory" to find out the treatment made with the last broadcast packet.

6. In both cases, whether the "last broadcast" packet was forwarded or discarded, the FCEE algorithm will take a decision according to the fuzzy rules presented in Table 2.

7. The output is calculated by using fuzzy rules, and a decision is made accordingly.

## SIMULATION MODEL AND RESULTS

The current section is concerned with the methodology used for this study. The presented article uses a simulation approach and parameters configured to execute the work successfully. Overall, the study highlights the need for network simulator 2 (NS-2) to perform the simulation. NS-2 is an open-source discrete event simulator widely used in research. The core engine of NS-2 is built on the C++ programming language, and the front system, which is used to create simulation topology, uses the Object-Oriented Tool command language (OTCL). NS2 is an event-driven simulation program beneficial in researching the dynamic nature of communication networks. The current work used the latest version of NS-2, which is 2.35. It generates two types of trace files, which are (i) simulation trace and (ii) nam trace. The simulation trace file is further used for data analysis.

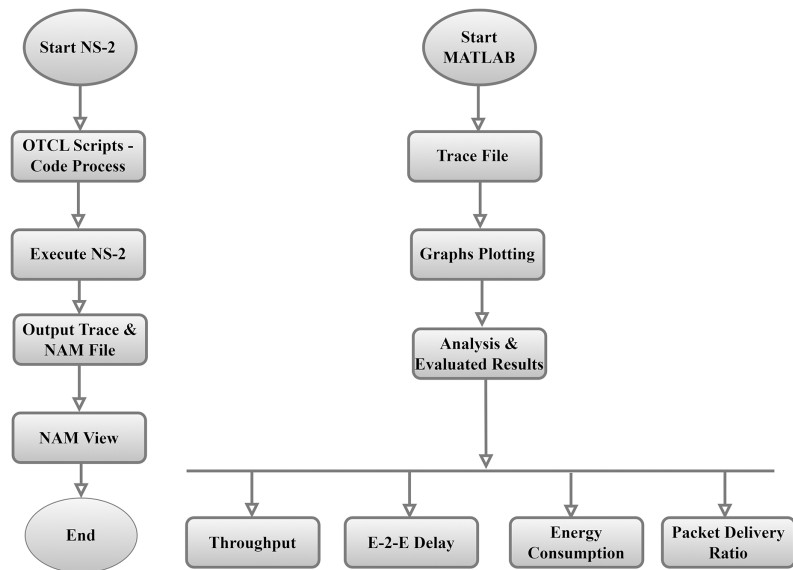

**Figure 9 NS-2 structure & code operations.**

In contrast, the Nam trace file can be fed into the network animator (Nam) utility to view how the simulation is carried out. Figure 9, shows how the simulation is carried out using NS-2. The MATLAB programming language has been used to generate the graphics and analyze the trace file by formulas and MATLAB.

In addition, the current work proposed a custom Perl script to calculate metrics such as packet drop rate (PDR), throughput, average end-to-end delay, and network overhead from the trace files. In addition, the scripts presented in OTcl, an object-oriented language-enhanced version of TCL for modelling and evaluating UDP protocols, routers, and other network items, execute the NS-2 software. Network scenario simulations were developed with TCL scripts, with comparable implementations for connection settings, node movement, and location. Other modifications were implemented to adjust the nodes' transmitting and receiving power to produce an effective influence per packet. The simulation study results are produced in a trace file that is included in the stimulation details of the network. The effectiveness of the proposed protocol is compared to that of widely used AODV variant protocols.

Table 3, displays the results of the simulations. The efficiency of the protocols is evaluated by looking at throughput, packet delivery ratio, end-to-end delay, and percentage of node survival. At the outset, the simulation area has a random distribution of nodes. Radio communications can be sent and received in any direction using an omnidirectional antenna ranging from 250 m per node. The quickest speed at which the nodes may travel is 10 m per second. Current work employs the Two-Ray Ground Reflection Model as a model for radio wave propagation. The simulation's traffic sources begin operating at the beginning and continue operating throughout. The source produces continuous bit rate (CBR) traffic. It creates UDP packets. Each datagram packet is 512 bytes long.

**Table 3 Simulation parameters.**

| Parameter | Value |
| --- | --- |
| Simulator | NS2 (version 2.35) |
| Routing protocols | FCEE, AODV & IRAODV |
| Topology area | 1,000 × 1,000 (meters) |
| Simulation time | 300 s |
| Mobility model | Random way point (RWP) |
| Traffic type | CBR |
| Speed (movements) | 10 m/s |
| No. of nodes | 100 |
| Radio propagation model | Two ray ground reflection |
| Minimum speed | 1 m/s |
| Pause time | 1 m/s |

To better reflect the real world's dynamic structure, the simulation's stop period has been set to 1 s. In the current configuration, the power usage of the node is set to 1.15 W while not in use (idle), 1.2 W when sending, and 1.6 W when receiving. Setdest was used to produce random motion according to the RWP model, and cbrgen was utilized to construct a random data transmission scenario. Primary importance is put on performance analysis in the context of mobility. Movement patterns from real-world mobile entities are hard to come by; hence, synthetic mobility models that approximate their behaviour are used instead. With this in mind, simple inferences concerning the most critical network characteristics from the model are provided. The mobility model characterizes the paths taken by the mobile nodes in the simulated experiment. Given that a routing protocol may perform excellently in one mobility model even while its performance in other conditions is unnecessary, this fact plays a key role in building and executing an excellent wireless infrastructure.

In most cases, the geographical extent of network nodes that are moving by the RWP model is not uniform. The rate at which each node travels towards the target location follows a normal distribution with a mean of zero meters per second and a maximum of the allowed speed. The RWP mobility model is utilized in this simulation because it is comparable to the movement pattern of users who are moving while carrying mobile phones and other devices, contributing to forming a wireless mesh network.

## Evaluation of parameters

These rather interesting evaluation results could be due to their relation to critical criteria in MANET networks; as described in the following text. The current work studied several parameters to gauge each routing protocol's effectiveness and performance. So, the proposed work will investigate the behaviour of Fuzzy Control Energy Efficient (FCEE) suggested routing protocols on evaluation parameters, which are presented below.

**Network throughput:** Network throughput relates to the number of packets successfully delivered to the receiver. All those packets which are lost due to any reason

affect the throughput of the network. Network throughput is measured in Kilobits per second (Kbps). Mathematically it can be expressed *via* Eq. (4).

$$Throughput = \frac{(Bytes\ Received) * 8}{(Simulation\ Time) * 1024} Kbps \tag{4}$$

**Packet delivery ratio (PDR):** The packet delivery ratio is the ratio of the total amount of packets (data) received by receiving nodes to the total number of packets created or produced in the network by source nodes. It can be expressed mathematically *via* Eq. (5).

$$PDR = \frac{(Packets\ Received\ (in\ numbers))}{Packets\ Sent} \tag{5}$$

**Average end-to-end delay:** The average end-to-end latency is the typical amount of time (in ms) it takes for a data packet to transit from its originator to its receiver across a network. A mathematical definition for an average end-to-end delay is shown in Eq. (6).

$$Average\ End\ to\ End\ Delay = \frac{1}{N}\sum_{i=1}^{N}(t_r - t_s)\ ms \tag{6}$$

where, $N$ = No. of data packets received, $t_r$ = timestamp at which $i^{th}$ packet is received, $t_s$ = timestamp at which $i^{th}$ packet was sent.

**Average energy consumption:** Determining MANET's energy consumption is possible by applying the Eq. (7). The computation of energy consumption for each node begins with subtraction: first, the initial value (i) of energy at each node is subtracted from the residual value (r) of energy; then, the value of nodes is divided by the total number of nodes in the network (N).

$$Average\ Energy\ Consumption = \frac{i - r}{N}. \tag{7}$$

## Results and discussion

This subsection presents the simulation results and the discussion of the results. In Fig. 10, a network throughput study is given for the FCEE, and Intelligent Routing protocol for Ad-Hoc On-Demand Distance Vector (IRAODV) are both routing protocols used in mesh.

FCEE has a higher network throughput of 352 kbps than IRAODV's 90 kbps, and 39 kbps for AODV. This means that FCEE can transfer more data in a given amount of time than IRAODV and AODV. The reason for achieving a higher throughput rate is introducing a memory channel, which controls and reduces the broadcast packets. Broadcast packets require and consume tremendous network bandwidth, and nodes, by default, can not avoid the processing of broadcast packets. Therefore, by limiting the number of broadcast packets circulating in the network, the suggested protocol achieves higher throughput rates.

This can be an essential factor to consider when choosing a routing protocol for a mesh network, as it will affect the network's overall performance. However, it is essential to note

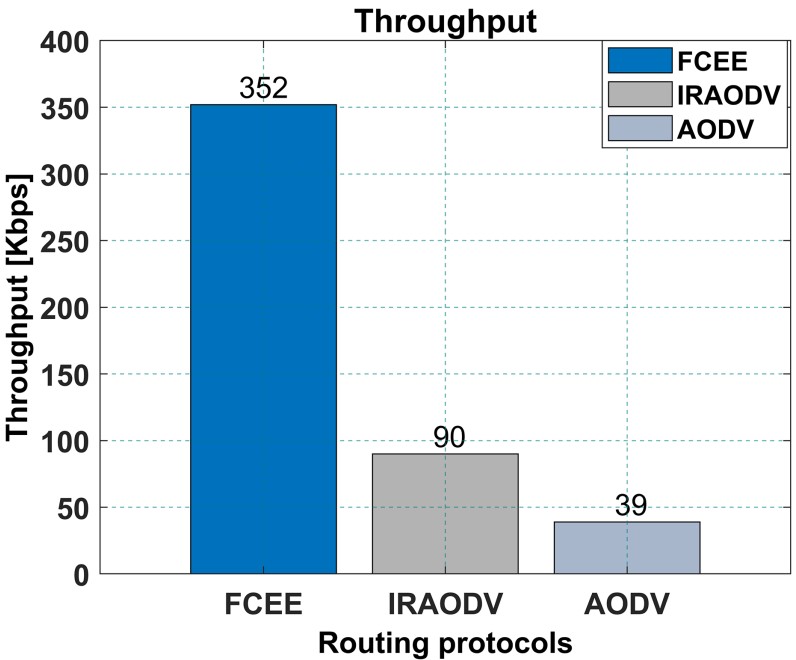

**Figure 10 Network throughput.**

that network throughput is not the only factor to consider when choosing a routing protocol. Other factors, such as end-to-end delay (E-2-E delay), packet delivery ratio (PDR), and energy consumption (EC), should also be taken into account. The article that is being presented is going to calculate and analyze all those factors. Overall, FCEE might be a better choice for a mesh network that requires high throughput.

The packet delivery ratio is a technique to understand the number of packets dropped. The higher number of packets delivered means fewer packets are dropped. Figure 11, presents packet delivery ratio results. FCEE has a higher packet delivery ratio of 98% compared to IRAODV's 68%, and the PDR for AODV was 35%. This means that FCEE is able to deliver more data packets to their intended destination than IRAODV. This can be an essential factor to consider when choosing a routing protocol for a mesh network, as it will affect the overall reliability and performance of the network. In the same context, this reduced percentage of dropped packets, which was about 2 per cent, shows promising results and makes FCEE suitable for heavily loaded networks.

Similarly, FCEE has a lower end-to-end delay of 13 ms compared to IRAODV's 55 ms, and 39 ms for AODV. This means that FCEE is able to transfer data from source to destination faster than IRAODV. This can be an important factor to consider when choosing a routing protocol for a mesh network, as it will affect the overall performance of the network, particularly for real-time applications such as voice and video communication, where a low end-to-end delay is crucial. FCEE might be a better choice for a mesh network that requires low end-to-end delay. Figure 12, presents the E-2-E delay in this particular network.

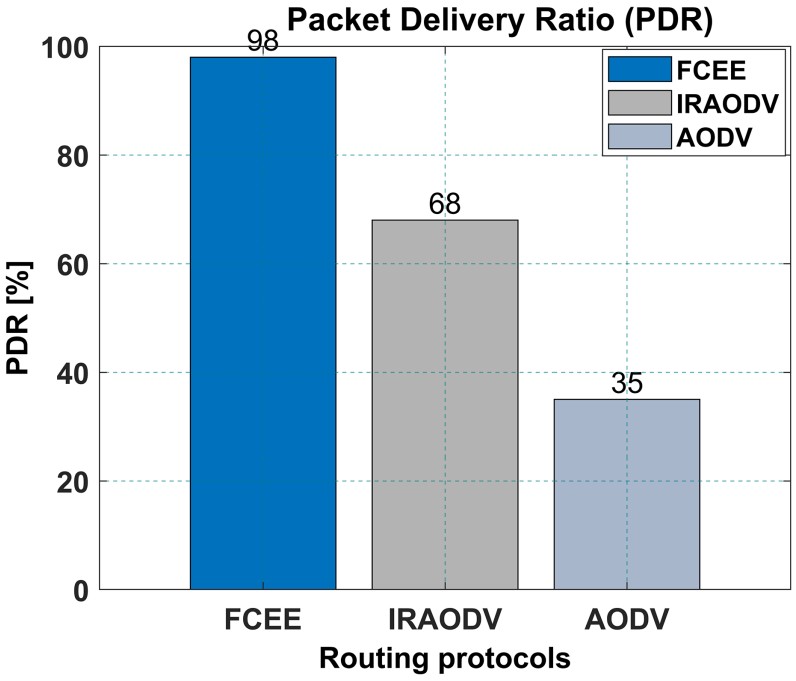

**Figure 11  Packet delivery ratio.**

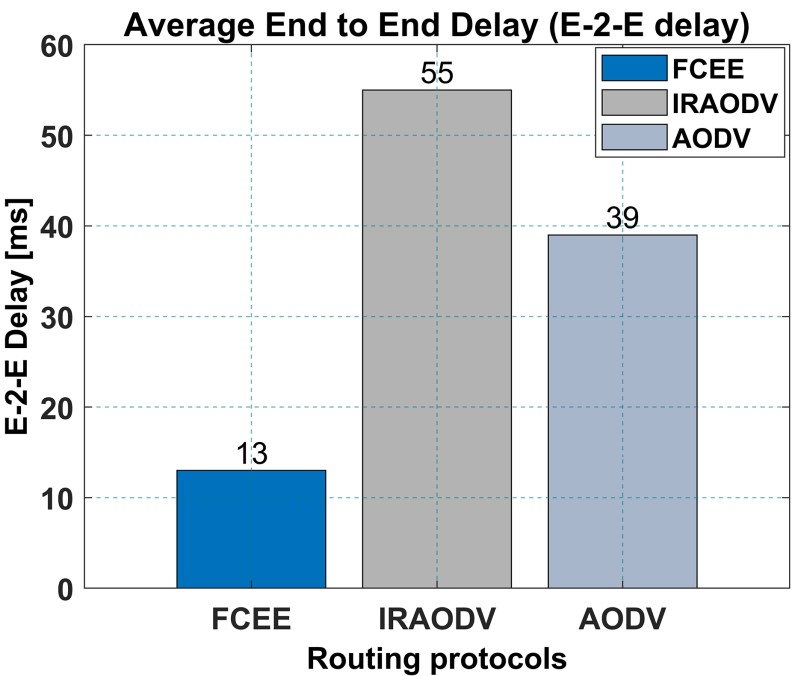

**Figure 12  Average end to end delay.**

The energy consumed by a routing protocol is an essential factor to consider when choosing a protocol for a mesh network, as it directly impacts the lifetime of the network's nodes.

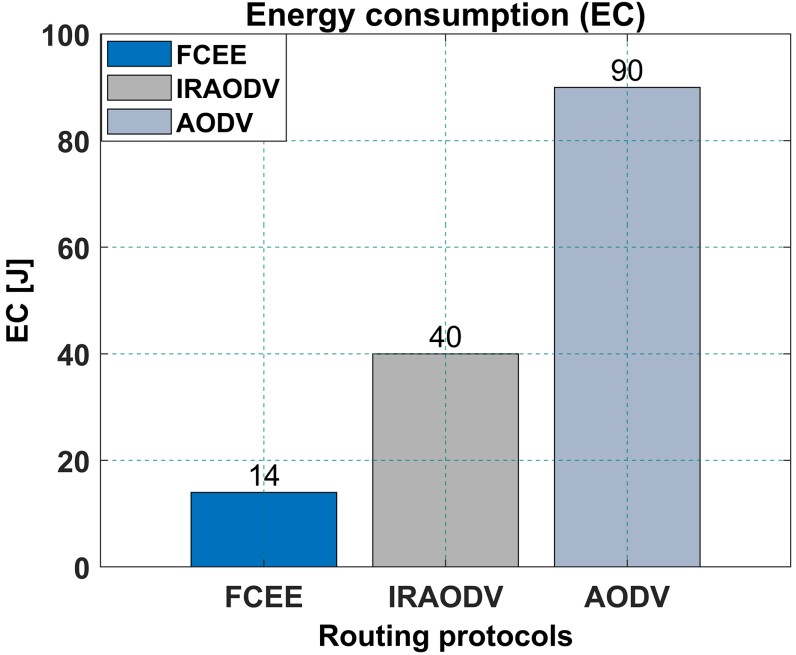

**Figure 13  Energy consumption.**               

In this case, it was observed that FCEE consumed 14 joules while IRAODV consumed 40 joules, while the AODV consumed 90 Jouls as shown in Fig. 13. This indicates that FCEE is more energy efficient than IRAODV. The lower energy consumption of FCEE can be attributed to its routing mechanism. In terms of results, this means that using FCEE instead of IRAODV can lead to an increased lifetime of the network's nodes. This can be particularly beneficial for applications that require the network to operate for extended periods in resource-constrained environments. Regarding energy consumption, as a reactive routing protocol, FCEE may have an advantage over routing protocols like IRAODV. Additionally, using fuzzy logic in the routing decision-making process may also help reduce energy consumption.

In conclusion, FCEE's reactive routing approach and fuzzy logic can lead to reduced routing overhead and energy consumption, which can be beneficial for mesh networks that require low energy consumption. In conclusion, FCEE's lower energy consumption than IRAODV makes it a more suitable choice for mesh networks that require extended network lifetime. However, as with any routing protocol, it is essential to consider all factors and evaluate the specific needs of the network before making a decision.

## CONCLUSION

This research article introduces a novel routing approach known as the Fuzzy Control Energy Efficient (FCEE) designed specifically for wireless mesh networks. The FCEE routing protocol harnesses the power of fuzzy logic, a versatile and promising concept extensively explored in various scientific domains. By judiciously evaluating the energy levels of network nodes and leveraging memory channels for informed decision-making,

the FCEE approach adeptly manages and restricts the transmission of broadcast packets within the network.

Compared to IRAODV, a reactive routing protocol, FCEE exhibits remarkable performance advantages. It achieves a significantly reduced end-to-end delay of merely 13 ms, whereas IRAODV requires 55 ms and AODV necessitates 39 ms. Moreover, FCEE achieves an impressive packet delivery ratio (PDR) of 98%, surpassing IRAODV's 68% and AODV's 35% PDR. Notably, FCEE achieves these exceptional outcomes while consuming a mere 14 joules of energy, a considerable improvement over IRAODV's 40 joules and AODV's 90 joules. The incorporation of fuzzy logic in the routing decision-making process contributes to enhanced network performance and adaptability.

In light of these compelling findings, FCEE emerges as an optimal choice for mesh networks that prioritize low end-to-end delay, high packet delivery ratio, low energy consumption, substantial throughput, and adaptability to evolving network conditions. Nevertheless, for further advancement in this field, it is advisable to conduct thorough evaluations of FCEE's performance across diverse environmental settings and varying traffic patterns. Additionally, a comparative analysis of FCEE with other reactive routing protocols could shed light on the trade-offs associated with different routing approaches.

In conclusion, the FCEE presents a potent solution for enhancing the efficiency of wireless mesh networks. Its successful application not only demonstrates the capabilities of fuzzy logic but also reinforces the potential for leveraging intelligent decision-making techniques to overcome the challenges faced by modern networks.

# ACKNOWLEDGEMENTS

We would like to express our sincere gratitude to the reviewers for their valuable feedback and insights on our manuscript. Their comments and suggestions have greatly helped to improve the overall quality of our work. We appreciate their time and effort in evaluating our research and providing us with constructive criticism. Thank you again for your invaluable contributions.

## Funding

This work was supported by the GS University of Pardubice project No. SGS_2023_013. The funders had no role in study design, data collection and analysis, decision to publish, or preparation of the manuscript.

## Grant Disclosures

The following grant information was disclosed by the authors:
GS University of Pardubice: SGS_2023_013.

## Competing Interests

Tawfik Al-Hadhrami is an Academic Editor for PeerJ.

## Author Contributions

- Ibrahim Alameri conceived and designed the experiments, performed the experiments, analyzed the data, performed the computation work, prepared figures and/or tables, authored or reviewed drafts of the article, and approved the final draft.
- Jitka Komarkova conceived and designed the experiments, performed the experiments, analyzed the data, performed the computation work, prepared figures and/or tables, authored or reviewed drafts of the article, and approved the final draft.
- Tawfik Al-Hadhrami conceived and designed the experiments, performed the experiments, analyzed the data, performed the computation work, prepared figures and/or tables, authored or reviewed drafts of the article, and approved the final draft.

## Data Availability

The code is available in the Supplemental Files.

## Supplemental Information

Supplemental information for this article can be found online at http://dx.doi.org/10.7717/peerj-cs.1508#supplemental-information.

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
