# Peer review of "Fuzzy-based optimization of AODV routing for efficient route in wireless mesh networks"

_PeerJ Computer Science, doi:10.7717/peerj-cs.1508_

## Round 0.1 · original submission · Minor Revisions

Based on reviewers’ comments, you may resubmit the revised manuscript for further consideration. Please consider the reviewers’ comments carefully and submit a list of responses to the comments along with the revised manuscript.

Reviewer 1 ·

Basic reporting

no comment

Experimental design

no comment

Validity of the findings

no comment

Additional comments

no comment

Annotated reviews are not available for download in order to protect the identity of reviewers who chose to remain anonymous.

·

Basic reporting

I have added all the review comments in the field for additional comments. The topic of the paper is relevant to the journal.

Experimental design

The Methods section of an experimental paper design details precisely what the research entails. But there are other comments, which are written in the comments below.

Validity of the findings

The authors claimed that their protocol had a good result compared with the standard one, where all analyses and comparisons performed on the data are reported in the paper.

Additional comments

Because of the widespread interest in the topic, conducting research on it in depth could prove to be both necessary and beneficial. On the other hand, the following modifications need to be taken into consideration:
Comments: -
1. The paper is written very well, and the text is easy to understand and not confusing at all.
2. I suggest enhancing figure number 5, and in line 337, the word " figure" appeared twice.
3. A typo with " FCEA".
4. The paper is Partially technically sound and needs to enhance.
5. The flowchart has many typos and needs to enhance.
6. Equation number 1 does not match with the text.
7. Was the comparison made with one protocol? Why?
8. This statement, " When a node receives a packet from the wireless channel, the Layer II handling and processing is performed by the data link layer while the network stack, e.g., TCP/IP, handles the packet and performs Layer III and IV operations. For example, calculation and verification of the header checksum, verifying the TTL field, etc. Layer IV generally checks and detects errors in the payload" this is not clear and needs to enhance.
9. FCEA has appeared in many places and has typos.
10. The manuscript presents a proposed routing protocol for a wireless mesh network that focuses on reducing energy consumption and improving network lifetime.

---

## Round 0.2 · accepted · Accept

Congratulations, the reviewer has expressed satisfaction with the revisions of the manuscript.

Reviewer 1 ·

Basic reporting

no comment

Experimental design

no comment

Validity of the findings

no comment

Additional comments

no comment

Annotated reviews are not available for download in order to protect the identity of reviewers who chose to remain anonymous.